# An Agent-Based Model for Disease Epidemics in Greece

**Vasileios Thomopoulos *** and **Kostas Tsichlas**

Department of Computer Engineering and Informatics, University of Patras, GR26504 Rion, Achaia, Greece; ktsichlas@ceid.upatras.gr
* Correspondence: vthomopoulos@upatras.gr

**Abstract:** In this research, we present the first steps toward developing a data-driven agent-based model (ABM) specifically designed for simulating infectious disease dynamics in Greece. Amidst the ongoing COVID-19 pandemic caused by SARS-CoV-2, this research holds significant importance as it can offer valuable insights into disease transmission patterns and assist in devising effective intervention strategies. To the best of our knowledge, no similar study has been conducted in Greece. We constructed a prototype ABM that utilizes publicly accessible data to accurately represent the complex interactions and dynamics of disease spread in the Greek population. By incorporating demographic information and behavioral patterns, our model captures the specific characteristics of Greece, enabling accurate and context-specific simulations. By using our proposed ABM, we aim to assist policymakers in making informed decisions regarding disease control and prevention. Through the use of simulations, policymakers have the opportunity to explore different scenarios and predict the possible results of various intervention measures. These may include strategies like testing approaches, contact tracing, vaccination campaigns, and social distancing measures. Through these simulations, policymakers can assess the effectiveness and feasibility of these interventions, leading to the development of well-informed strategies aimed at reducing the impact of infectious diseases on the Greek population. This study is an initial exploration toward understanding disease transmission patterns and a first step towards formulating effective intervention strategies for Greece.

**Keywords:** agent-based modeling; COVID-19; SEIR model; synthetic population; disease dynamics

## 1. Introduction

During the COVID-19 pandemic peaks in Greece, there was a daily update by a scientific committee regarding the number of infections and other related statistical information. Based on this information, the scientists were arguing in favor of specific interventions that would help mitigate the large wave of infections. Their prime argument related to the proposed interventions was the predictions based on models about the progress of the pandemic if left unchecked. The question was raised about what models they were using to come to such decisions. It turns out, and to the best of our knowledge, that the models used by the scientific committee in Greece were based on an ODE approach. It is worth noting that while the alternative ABM approach has been utilized in other countries, there has not been a study applying this approach specifically to Greece. Generally, and in the case of epidemiology particularly, scientists use various tools, among which are predictive models, in order to promptly react to outbreaks as well as to understand the disease dynamics.

Simulation is a major tool for studying disease spread and it may be based either on an approach centered on ordinary differential equations (ODE) (a descriptive approach) or Agent-Based Modeling (ABM) (a mechanistic approach). In the ODE approach, the population is divided into compartments (groups) and the aggregate number of individuals that have a particular state (e.g., infected) is a function of time. These equations essentially describe the volume of people that move between different states (compartments) as a function of time. They can also be used to compute various characteristics of the population with respect to the epidemic, like the basic reproductive number. Conversely, agent-based

models simulate the population by entities (of different granularities depending on the application), corresponding to agents with specific traits and behaviors. In the case of epidemiology, each agent corresponds to a human within the population. These agents interact at a local level, and a global behavior emerges (e.g., an outbreak) as a result of these local interactions, while the ODE (aggregated) and ABM (disaggregated) methods both have their strengths and weaknesses, one notable strength of ABM is the fine-grained opportunities it offers to change agent traits, behaviors, interactions, and disease parameters while looking at their effects on global behavior. Interventions and sensitivity analyses are two examples that make use of such refinements. See the study by Bjørnstad et al. [1] for a starting point for discussing such models. Indeed, one of the most important practical results and major impact of ABM in epidemiology is policy recommendation. There are quite a few studies related to how interventions affect the progress of an outbreak [2] as well as the economic impact of such interventions [3].

In this paper, we present the first steps toward developing an ABM tailored to the unique characteristics of the Greek territory. The development of an ABM system for epidemics in Greece poses particular challenges due to a unique combination of social and geographical features (as, of course, is the case for any other country/region). This ABM must take into account the geographical distribution of the Greek population like the numerous populated islands, its capital with almost half of the country's population, many small villages on its mountainous terrain, etc. Similarly, it must also take into account social interactions that involve religious activities, which are quite intense among the elderly or during specific time periods (e.g., Orthodox Easter), outdoor activities that are also quite intense during summer (e.g., folk festivals), the frequent travel of people from the big cities to their family homes in villages, etc. Finally, family relations are quite strong in Greece, and this is one of the reasons why many families contain three generations (children, parents, grandparents) living in the same house, or the children tend to leave home at a relatively older age (after 25) due to financial difficulties. Our long-term goal is to assist policymakers in implementing effective intervention measures through the exploration of various what-if scenarios. The model will provide a fine-grained understanding of disease spread, giving access to valuable insights for decision makers. Our contribution towards this goal is along two axes:

1. We present a framework for creating synthetic populations in the Greek territory, utilizing statistical distributions derived from census data from various public sources. These synthetic populations serve as the foundation for a data-driven, multi-layer agent-based model specifically designed to simulate the dynamics of infectious diseases in Greece;

2. We apply an SEIR virus propagation model to simulate the dynamics of the virus on the synthetic population.

The rest of the paper is organized as follows. In Section 2, we discuss basic notions used throughout the paper and also provide an overview of the relevant literature. In Section 3, we describe the methodology used throughout the paper. In particular, we analyze our approach for creating the synthetic population and we also discuss the virus propagation model. The experimental results and related discussion are given in Section 4, and we conclude in Section 5.

## 2. Preliminaries

*Agent-based models* (ABMs) are microscale models that simulate multiple agents' simultaneous operations and interactions in an attempt to re-create and predict the appearance of complex phenomena. In ABMs, higher-level system properties arise from the lower-level subsystem interactions, a phenomenon known as *emergence*. ABMs have been used extensively for modeling purposes in non-computing related scientific domains like biology, ecology, and social sciences [4].

Traditional epidemiological research focuses on rate-based differential equation modeling of perfectly mixed populations, in the sense that each agent can interact with any

other in the population. This approach allows analytical expressions to be obtained for certain quantities like the number of infected people in a population [5]. However, this is true for simple models only, in the sense that the number of population groups is small. In addition, such an approach imposes very strong assumptions about one of the critical factors in disease evolution: human interaction (assuming perfect mixing). To this end, Agent-Based Modeling (ABM) can be used as an implementation approach for complex disease models as well as rich human interactions. ABM provides the modeler with an extensive toolkit with respect to checking policy planning and making detailed predictions. The main hurdles related to this approach are the need for high-quality data and the difficulty in parameterization and tuning of the model, as well as efficiency issues that come into play when the population is large.

Applying the Agent-Based-Modeling paradigm to epidemics requires two major components: the population and the disease model. Epidemiological Agent-Based Modeling relies critically on the definition of a population that is usually synthetic. The former entails statistical summaries or distributions of various population characteristics, often represented in the form of statistical tables, such as the count of individuals within specific age groups or income brackets. Conversely, disaggregated data consists of individual-level records for persons or households, encompassing diverse attributes like age, income, gender, and more.

Synthetic populations, generated using computer simulation techniques, are statistical representations of actual populations. They play a vital role in research and policy analysis. They provide valuable insights into the behavior and characteristics of populations while also allowing for the evaluation of different policies and interventions on population outcomes. To create a synthetic population, various attributes like age, gender, race, education, occupation, income, and location are carefully specified. Statistical models and algorithms are then used to simulate these attributes, either by utilizing real-world data or by making assumptions about their distribution within the population.

In particular, a synthetic population requires the definition of the agents and their properties, their spatial placement, and finally their relations. In fact, this is usually the order in which the complete synthetic population is generated. Traditionally, population synthesis can be achieved through three approaches [6]: (1) Synthetic Reconstruction (SR) (e.g., [7]), (2) Combinatorial Optimization (e.g., [8]) (CO), and (3) Statistical Learning (SL) (e.g., [9,10]). Recently, there has been a very small number of studies that adopt machine learning (ML) approaches (e.g., [11]) that fall under the umbrella of SL but support efficiently the generation of synthetic populations with a high number of characteristics/traits, which is certainly not the focus of this paper. In principle, both SR and SL are based on samples of the population, although, SR methods have been devised that are sample-free. Sample-free methods try to rebuild disaggregated population data from the aggregated real population data, while sample-based methods try to generate the entire population by replicating the available disaggregated data, which is a sample of the real population. In Lenormand and Deffuant [12], a comparison is made between a sample-free method and a similar sample-based method leading to the conclusion that the sample-free method is superior regarding the population fitting although it requires more preprocessing. In Yaméogo et al. [6], a comparison is given between these approaches for generating a two-layered population (individuals and households), while at the same time, they propose a decision-making procedure as to the best approach available based on the characteristics of the data that describe the real population.

Initially, the synthetic agents and their properties need to be defined, resembling the target population with respect to various statistical measures (e.g., age distribution). The key objective is to reduce the disparity between the synthetic population and the actual population concerning these statistical measures.

The social interactions between the agents in the population are usually represented by networks. Usually, *multi-layer networks* are used that are able to characterize numerous types of interactions that a typical monolayer network approach cannot. Edges (or links) in

monolayer networks describe undirected or directed connections between nodes. Networks might, for example, describe social relationships (undirected edges) among wild animals (each individual being a node) or movement (directed edge) from one farm to another (each farm being a node). Multilayer networks likewise have nodes and edges, but the nodes are separated into layers, each indicating a particular type of interaction that connects to form an aspect. Aspects, or layers stacked together, can be used to represent many sorts of contacts, physical locations, subsystems, or temporal points. Intralayer connections are made between nodes in the same layer of an aspect, whereas interlayer connections are made between nodes in different layers [13,14]. Each layer can be a different type of a random network depending on the degree distribution, and as such, the appropriate generation process must be adopted. Non-random networks are rarely used and only in cases where the full network can be reconstructed from the available data. The type of random network depends on the relation type. For example, modeling households corresponds to a network that consists of a set of small cliques (the size of the family) whose distribution is a power law. Conversely, modeling random interactions in a geographical area can be simulated by a random network that corresponds to homogenous mixing.

*Related Work*

Before discussing ABM-based systems for virus propagation that contain a population synthesis module, we briefly discuss open-source approaches (we do not consider approaches that have not published their code) for population synthesis in general. The creation of synthetic populations (or ecosystems to be more precise) begins all the way back in 1996 with the Transportation Analysis Simulation System (TRANSIMS) [15], which simulates traffic patterns on synthetic individuals. In a rather outdated survey on population synthesizers [16], the goal was set to create a general software solution. Indeed, in 2017 a general framework for generating Synthetic Populations and Ecosystems of the World (SPEW) was implemented [17]. SPEW supports various sampling methods for constructing the synthetic population and their location within a geographic region. Given that appropriate data exist, SPEW can create a synthetic ecosystem for different agents. The authors also claim to have generated a synthetic population for over 70 countries worldwide (among which is Greece) based on the data taken from Integrated Public Use Microdata Series International (IPUMS-I) [18]. However, the web domain is deactivated and this synthesized data cannot be found anywhere. Even if the data were available, the granularity level of the synthetic population would not be enough for our needs.

Understanding the current state of medical knowledge about COVID-19 and considering demographic factors are crucial when developing strategies to mitigate its spread. Simulation models can aid policymakers in making informed decisions by taking into account the prevailing situation. Agent-based modeling (ABM), which incorporates human behavior and interactions, has proven particularly useful in studying the spread of COVID-19. A comprehensive literature search yielded several papers that focus on agent-based modeling of COVID-19 transmission pathways (a representative small subset is shown in Table 1 and discussed further below). These models vary in their objectives, the number of simulated individuals (agents), the geographical areas they represent, and their approaches to modeling transmission patterns, illness states, human behavior, and treatments. However, a gap exists between policymakers' requirements and the capabilities of simulation models in accurately reflecting real-world factors that influence human decision making and transmission dynamics [19].

Chang et al. [20] report on the results of agent-based modeling of the COVID-19 outbreak in **Australia** using a fine-grained computational simulation. This model has been calibrated to meet important COVID-19 transmission parameters. The age-dependent fraction of symptomatic instances is a key calibration outcome, with this fraction for children being one-fifth of that for adults. The model is used to compare a variety of intervention techniques, including international flight restrictions, case isolation, home quarantine, social separation with varied levels of compliance, and school closures. School

closures do not appear to provide significant benefits unless they are accompanied by a high level of social distancing compliance.

In the study by Hoertel et al. [21], a stochastic agent-based microsimulation model of the COVID-19 epidemic in **France** is presented. The model also assesses the possible effects of post-quarantine interventions, such as social isolation, mask usage, and population shielding of those most susceptible to severe COVID-19 infection on the disease's overall incidence and death, as well as on the utilization of ICU beds. The model was accurately calibrated, and changes in model parameter values had no effect on the estimations of the results. The authors concluded that even while quarantine is efficient in stopping viral transmission, no matter how long it lasts, it is unlikely to stop the epidemic from spreading again.

In the study by Hunter et al. [22], an agent-based model for towns in **Ireland** was created. A data-driven agent-based model to mimic the development of an airborne infectious illness in an Irish town using publicly accessible data was built. By recreating a measles outbreak that happened in Schull, Ireland in 2012, the model was put to the test. The same outbreak in 33 different towns was replicated, and then the relationships between the model's output and the attributes of each town (such as population, area, vaccination rates, and age distribution) were examined to see if these attributes have an impact on the model's output.

In the study by Canabarro et al. [23], a multi-layer network with an extended SIR disease model taking into account homes, transport, workplaces, schools, religious activities, and random encounters was considered for the COVID-19 virus in **Brazil**. Due to efficiency issues, the authors considered only a population of $10^5$ agents that matched the census data of Brazil and then scaled up their results. By explicitly calculating the demand for hospital ICUs in the case where the schools and universities are closed, social isolation of individuals over 60 is imposed and home quarantine is proposed on a voluntary basis, they show that the Brazilian health system cannot cope with the demand.

EnerPol is another agent-based simulation framework used in disease spread and utilized by Marini et al. [24] to study influenza and COVID-19 epidemics in **Switzerland**. The authors offer a stochastic model for daily activities as well as a mobility model and they also take into account mesoscopically the weather. All this information is extracted from publicly available data. The daily activities model generates the contacts when agents are in the same place (workplace, school, etc.), while the mobility model allows public/private transportation to be taken into account. For the mobility model, the road network and the train network, as well as any public transportation method, have been taken into account (e.g., buses or airports). For approximately 9 million agents over a 3-month period with sub-hourly time steps, a single scenario requires a few hours to run on a GPU.

A less sophisticated model called REINA (2020) (Realistic Epidemic Interaction Network Agent) [25] maintains 1.6 million agents in the region of Helsinki University Hospital, **Finland**. The implementation is open-source and one instance runs within a few seconds on their online platform. The agents are individuals with certain properties while the epidemic model is basically an SEIR with additional states related to hospitalization or stays in an ICU. The contact network is rather simple: each day, an agent according to some age-dependent distribution has certain random contacts. Thus, the contacts are basically random (homogenous population mixing) and there is no consideration of modeling real social networks (e.g., work environments).

FRED (Framework for Reconstructing Epidemic Dynamics) is tailored for epidemic diseases in the **USA**. FRED is an open-source agent-based modeling system that is free to use and closely based on models used in earlier studies of the pandemic flu. FRED makes use of open-access census-based synthetic populations that accurately reflect the demographic and geographic diversity of the population, as well as social networks in the workplace, in homes, and in schools. Every state and county in the United States as well as a few other countries presently have access to FRED epidemic models [26].

In the study by degli Atti et al. [27], the authors use an SEIR model for case importation and an individual-based model (IBM) for modeling the spread of pandemic influenza in **Italy**. The impact of various control strategies was assessed. Travel destinations that matched the information from the 2001 census for the 57 million Italians were used. Several $R_0$ values (1.4, 1.7, and 2) to assess the effect of control methods (vaccination, antiviral prophylaxis, international air travel restrictions, and increased social distancing) were used.

In the study by Ng et al. [28], an age-structured agent-based model of the **Canadian** population was created to simulate how public health actions at present and anticipated levels may affect the spread of SARS-CoV-2. Case identification and isolation, contact tracing and quarantine, physical seclusion, and community closures were among the interventions that were tested separately and in combination.

In the study by Koh et al. [29], the authors simulate the spread of COVID-19 Omicron by using an innovative three-dimensional agent-based model that takes into account **Hong Kong**'s vertically extended hyperdense urban environment. The model evaluated the efficacy of the "zero-COVID" initiatives, such as citywide lockdown and mandatory universal testing (CUT), that were under discussion during the Omicron wave in Hong Kong. It was discovered that even quicker and tougher execution was required for such rigorous interventions to be successful. They conclude that adaptable long-term methods for controlling and preventing future epidemics should also be taken into consideration.

In the study by Ning et al. [30], the spread of COVID-19 among the 11.2 million residents of Shenzhen City, **China**, was replicated using a spatially explicit agent-based model. This was achieved by integrating huge mobile phone tracking records, census data, and building features. The model was used to determine the likelihood of a COVID-19 comeback if sporadic cases appeared in a city that had been entirely restored after it had been validated by local epidemiological evidence. At different degrees of public compliance, combined scenarios of three crucial non-pharmaceutical treatments (contact tracking, mask-wearing, and quick testing) were evaluated.

In the study by Gharakhanlou and Hooshangi [31], an agent-based model is presented that replicates the spatiotemporal patterns of the COVID-19 epidemic. The effects of various COVID-19 outbreak control tactics, including office closures, social exclusion, and closing of schools and educational facilities, in Urmia City, **Iran**, are examined. All control methods used in Urmia City together with the accompanying actions of each control strategy were incorporated into the ABM. The transmission of COVID-19 between human agents was replicated using the SEIRD propagation model.

In the study by Gomez et al. [32], an agent-based model named INFEKTA is proposed for modeling the spread of infectious diseases subject to social distance regulations. INFEKTA combines demographic data (population density, age, and different types of people) from geographical regions of the actual town or city under investigation with the transmission dynamics of a particular disease (according to parameters discovered in the literature). Agents (virtual people) can roam through a complicated network of accessible venues defined on an Euclidean space that represents a town or city in accordance with their mobility patterns and the imposed social separation policy. With one million virtual people, INFEKTA simulates the COVID-19 transmission dynamics in Bogotá, the capital of **Colombia**, under various social exclusion policies. Based on the sensitivity study of the effects of social distance policies, the authors concluded that the implementation of "medium" strength social distance policies (i.e., closure of 40% of the sites) significantly reduces the spread of the disease.

In the study by Singh et al. [33], COVID-19 propagation modeling results for several mitigation and confinement scenarios are presented for the Madrid, **Spain**, metropolitan region. Utilizing EpiGraph, an epidemic model that has been enhanced to replicate COVID-19 spread, these scenarios were put into practice and tested. In order to create a social interaction model that accurately reflects a variety of individual and group traits as well as their unique linkages, EpiGraph analyzes connection patterns in social networks. Along with the epidemiological and social interaction components, a transportation model is used

to simulate how individuals move across short and large distances. These characteristics provide EpiGraph with the ability to replicate COVID-19 development and identify the medium-term consequences of the virus when using mitigation techniques, in addition to the ability to model scenarios with millions of people and apply various contention and mitigation mechanisms. In the Madrid metropolitan region, EpiGraph achieves closely linked infection and death curves associated with the first wave, attaining comparable seroprevalence levels. The authors demonstrate the reduction of the mortality toll when a selective lockdown policy for the elderly (over 60) is imposed. Additionally, the impact of mask-wearing after the initial wave was considered, demonstrating that a key element in limiting the spread of the virus is the proportion of people who wear masks as instructed.

In the study by Bicher et al. [34], a method for calculating the level of immunization in the Austrian population and a discussion about possible repercussions on the effects of herd immunity were discussed. A calibrated agent-based simulation model that accurately simulates the COVID-19 epidemic in **Austria** is used to determine vaccination rates. The number of vaccinated individuals may be determined from the generated synthetic individual-based statistics. Then, by altering the acquired degree of vaccination in simulations of an imagined uncontrolled epidemic wave, the pandemic's course to show potential implications on the effective reproduction rate was extrapolated.

In the study by Xu et al. [35], an iterative process based on data from land use, questionnaire surveys, and population censuses was used to create a synthetic population for **American Samoa**. The population serves as the foundation for an agent-based model created primarily to close knowledge gaps regarding the transmission and eradication of lymphatic filariasis while also being easily adaptable to mimic other infectious diseases. The statistically realistic population and household structure, as well as the high-resolution geographic placements of households, were characteristics of the synthetic population. From 2010 to 2050, the population was simulated over a 40-year period. The projected and estimated populations from the U.S. Census Bureau were contrasted with those from the simulation. The findings suggested that due to the huge number of emigrants that were seen, the total population would continue to decline. The study indicated that the population was aging, consistent with the estimates from the Bureau and the two latest population censuses. The examination of sex ratios across various age groups indicated a rise in the percentage of males in both the 0–14 and 15–64 age brackets. Concerning household size, the simulation consistently followed a Gaussian distribution, with an average size close to 5.0. Interestingly, this average size was slightly lower than the initial average size of 5.6.

In the study by Mossong et al. [36], a population-based prospective study on mixing patterns in eight European countries using a conventional paper-diary method was conducted. It was found that across many European countries, mixing patterns and contact features were remarkably similar. Strongly assortative age-related interaction patterns were seen, with young adults and schoolchildren in particular being more likely to associate with people of similar ages. Preliminary modeling predicts that during this measurement's initial epidemic phase, when the population is most vulnerable, children aged 5 to 19 will have the highest prevalence of a new virus disseminated by social contacts.

In the study by Rodríguez et al. [37], the researchers focus on developing a high-resolution, data-driven agent-based model to analyze the spread of COVID-19 in five Spanish cities: Barcelona, Valencia, Seville, Zaragoza, and Murcia. Utilizing synthetic populations based on multiple data sources, the model incorporates detailed interaction environments through multi-layer networks, considering home, nursing homes, school, work, university, and community layers. The research aims to simulate and assess the impact of various non-pharmaceutical interventions on COVID-19 transmission. The work addresses the need for quantitative approaches to characterize intervention impacts, which can vary based on cultural, regional, and population-specific circumstances. By presenting a detailed framework, the study offers a tool for simulating different intervention scenarios, contributing to evidence-based decision making in managing the pandemic. The

model's effectiveness is demonstrated through a case study, illustrating the impact of key interventions in the studied cities.

In the study by Peng et al. [38], the limitations of existing models at capturing COVID-19's impact on human mobility at a neighborhood level are addressed. Employing an agent-based model (VIABLE), the study simulates individual mobility choices based on social activities in neighborhoods, focusing on Porto Alegre, **Brazil**. The model considers agents' adaptation to exposure risks and their impact on well-being, revealing temporal variations and segregation in mobility patterns among agents with different vulnerability levels. The results highlight the shift in mobility choices during the pandemic, influenced by socio-demographic factors like age, car ownership, and economic status. While previous studies explored mobility tendencies at larger scales, this model aims to bridge the gap, providing insights into individual-level adaptations and neighborhood-specific mobility patterns under COVID-19, offering a nuanced perspective for urban planning and public health interventions.

In the study by Rykovanov et al. [39], the spread of a viral infection was modeled using agents representing citizens of the Moscow Oblast, **Russia**. In the study by Venkatramanan et al. [40], an agent-based model framework was created to predict the **Liberian** Ebola epidemic in 2014–2015 and then used for Ebola forecasting. GSAM [41] is a global-scale (billion agents) ABM Java framework. Its efficiency is critically based on the spatial homogeneity of the population at a specific granularity level. GSAM is an agent-based epidemic modeling high-performance distributed platform that can simulate a disease outbreak in a population of several billion agents.

Some general frameworks are not targeted to specific countries; these include the 'Oxford model' and Repast. The 'Oxford model,' often associated with the CovidSim framework developed by the Imperial College COVID-19 Response Team, has received notable publicity and it is indeed been widely used in various countries for modeling pandemic dynamics [42–44]. Additionally, Repast, a library/framework commonly utilized for ABM implementation, is a notable tool in this domain among others, although our study utilized the Mesa library [45].

There are numerous other relevant research works. In the study by Zhang et al. [46], the applications of three simulation approaches (System Dynamics Model—SDM, Agent-Based Model—ABM, and Discrete Event Simulation—DES) and their hybrids in COVID-19 research are systematically reviewed. Out of 372 eligible papers, 72 focused on COVID-19 transmission dynamics, 204 evaluated interventions, 29 predicted the pandemic, and 67 investigated the impacts of COVID-19. ABM was the most widely used simulation method (275 papers), followed by SDM (54 papers), DES (32 papers), and hybrid models (11 papers). The primary focus was on evaluating and designing intervention scenarios, accounting for 55% of the papers.

**Table 1.** Indicative Agent-Based Models comparison per country.

| Country | Population Creation | Number of Agents | Model Type | Infection Model | Year | Reference |
|---|---|---|---|---|---|---|
| Australia | census, national data sources | 0.5 m | several mixing groups | SEIR | 2020 | [20] |
| France | previous work, papers | 0.5 m extrapolated to 67 m | stochastic microsimulation ABM | not defined | 2020 | [21] |
| Ireland | census mainly | 0.1 m | NetLogo User Community Models | SEIR | 2018 | [22] |
| Brazil | census | 10 m | multi-layer network | SIRD | 2020 | [23] |
| Switzerland | synthetic population from census | 9 m | ABM and a stochastic model that simulates, on a sub-hourly timescale, the different daily activities of all individuals | not defined | 2020 | [24] |

**Table 1.** *Cont.*

| Country | Population Creation | Number of Agents | Model Type | Infection Model | Year | Reference |
|---|---|---|---|---|---|---|
| Finland | census statistics | 1.6 m | random interactions | SEIR | 2020 | [25] |
| USA | synthetic population from census | 30 m | mixing patterns | SEIRS | 2013 | [26] |
| Italy | census | 57 m | multi-layer network | SEIR | 2008 | [27] |
| Canada | projections | not defined | multi-layer network | SEIR | 2020 | [28] |
| Hong Kong | synthetic population from census | 0.73 m | three-dimensional vertically expanded | not defined | 2022 | [29] |
| Shenzen, China | mobile phone records, census | 11.2 m | spatially explicit ABM | SLIR | 2021 | [30] |
| Urmia, Iran | census and spatial data | 0.75 m | mobile & static agents | SEIRD | 2020 | [31] |
| Bogotá, Colombia | synthetic population from census | 9 m | random network | SEIRMD | 2021 | [32] |
| Madrid, Spain | census and social network data | 5 m | multi-layer network | SEIR | 2021 | [33] |
| Austria | census | 9 m | multi-layer network | not defined | 2022 | [34] |
| Moscow oblast, Russia | census | 10 m | multi-layer network | SLIR | 2022 | [39] |
| American Samoa | census, questionnaires and land usage | 0.055 m | age and household distribution, population evolution | not defined | 2017 | [35] |

## 3. Methodology

There are many challenges in developing ABM systems. First, an appropriate theoretical framework must be in place in order to describe and reason about this system. We choose to use graph dynamical systems [47], which model discrete dynamical processes on networks. This choice suits perfectly our case due to our intended use of multi-layer networks as the environment of the ABM. Second, the construction of the synthetic population requires the extensive use of data that come from multiple sources. These data may be incomplete, at different levels of granularity (e.g., large or small age groups), and may be contradictory. Data fusion under these conditions is quite challenging. Third, the number of agents, the number of scenarios, the number of total interactions, and the time granularity make unavoidable the use of parallelism. In addition, the storage of all these networks requires special care, since naive storage may be space-consuming and time inefficient with respect to retrieval. In this paper, we do not consider this challenge since we focus on a small area of Greece. Fourth, the sophistication of the disease propagation model drives the functionality of the system. Simple models like SIR are easy to apply and have only a few parameters but fail, however, to express many important states for epidemiology (like Asymptomatic). They also do not support sophisticated propagation models based on the characteristics of the particular virus (air-borne or contact-based). The propagation model must allow for building different scenarios based on interventions (e.g., obligatory masks in public spaces) or behavior modification (e.g., stay home when sick). In this paper, we employ the rather simple SEIR model.

Our approach is similar to that of [48]. The intended method for generating the synthetic population and its representation is depicted in Figure 1 (inspired by [49]). We discuss all these steps, although our results do not contain spatial information.

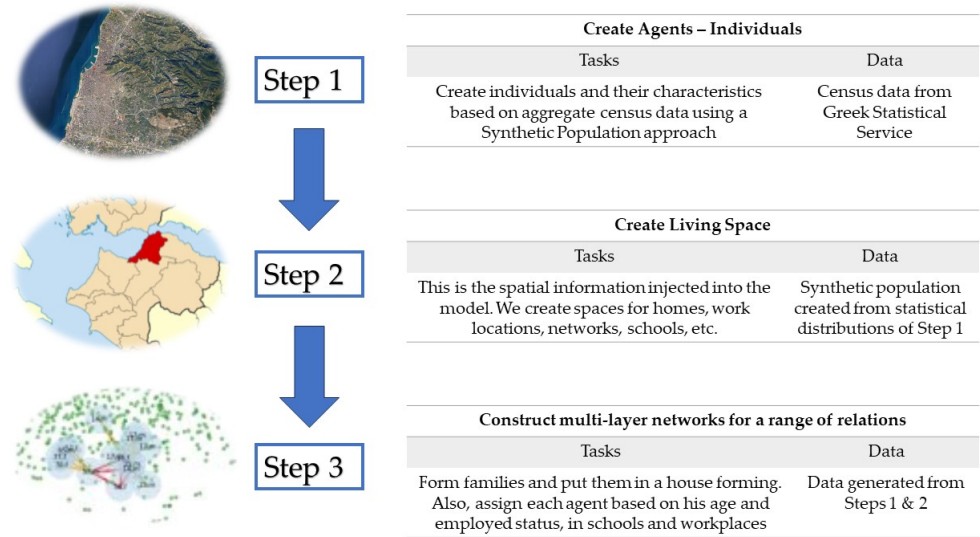

**Figure 1.** A high-level description of the generation process for the synthetic population.

1.  **Create Agents—Individuals:** Create individuals and their characteristics based on census data using an SR approach [50]. We adopt the SR approach since only aggregate data is available for Greece;
2.  **Create Living Space:** This is the spatial information injected into the model. We create space for representing relations within homes, work locations, and schools. We do not consider geographical information;
3.  **Place agents into groups:** Form families and other groups (work, school, etc.). To make a fine-grained synthetic household, we use an SR approach similar to the one in [51,52] and we further enhance it by using certain rules and heuristics that have been inferred from census data (e.g., [53]).

The characteristics of the population real data are used to generate multi-layer networks that define contacts or relations between agents. These layers have been constructed independently. We could not find sufficient data to determine the potential correlations between the layers, leading to the assumption that the layers are independent. This independence assumption is quite strong since there are statistical dependencies that should be taken into account. For example, we first generate individuals and then we place them in households. There are approaches that generate households with individuals in a single step, aiming at getting more realistic populations. We have defined the following layers:

1.  **Household:** A set of cliques for the members of a family. This is the easiest task considering that families have already been formed from the first step. This network represents relations and as such, it is static within a limited time horizon;
2.  **Work:** A set of small-world networks between agents in the same workplace. This is a contact network. An open-source method for creating home/work/school networks that follow this methodology can be found in [54];
3.  **Schools:** A set of small-world networks corresponding to the interactions between students;
4.  **Random-contact networks:** These correspond to random interactions between agents within the world. Small-world networks are used and they change between successive steps;
5.  **Friendship Networks:** Strong ties in the form of friendships are represented by a static network that could further give rise to more regular contacts. This network is formed by a social-mixing matrix that measures the frequency of relationships between agents in the same or different age groups. We have adopted the social-mixing matrix approach that is inferred by publicly available data of physical contacts and interactions for a country with a similar socio-economic structure and mentality, specifically, Italy [55]. Indeed, there are strong similarities between the two neighboring Mediterranean countries of Greece and Italy. These similarities include a common

history, shared cultural values, strong family ties, and a laid-back social atmosphere. Due to the unavailability of specific social interaction data for Greece, we utilized data from the Italian population, considering the numerous resemblances between the two nations. This practice is common in research, particularly when data for a specific region are not readily accessible, and a comparable region is considered representative. However, it is important to note that this approach is subject to future refinement as part of ongoing research to obtain dedicated social interaction data for Greece.

Concerning the propagation model, we adopt a simple SEIR (Susceptible–Exposed–Infected–Recovered) model with a small set of parameters.

The disease propagation model is executed in multiple scenarios. A scenario specifies the behavior of individuals as well as the applied public policies. Such scenarios are implemented by changing accordingly the layered networks while affecting the probabilities of the disease model.

### 3.1. The Synthetic Population

We create the synthetic population by obtaining data from publicly accessible sources. Demographics and census data are obtained from the Hellenic Statistical Authority (https://www.statistics.gr, accessed on 31 December 2023), other open data from the Ministry of Digital Governance (http://www.data.gov.gr, accessed on 31 December 2023), geographical data from open geodata (https://geodata.gov.gr, accessed on 31 December 2023) and workforce data from the Greek Manpower Employment Organization (https://www.dypa.gov.gr/statistika, accessed on 31 December 2023). We create a synthetic population mimicking the real attributes of the center of a major Greek city (Patras). We used real data (e.g., age and gender distribution) for the city of Patras from the Greek Statistic Service (ELSTAT) (https://www.statistics.gr/el/statistics/-/publication/SAM03/2011, accessed on 31 December 2023). We assume that all agents younger than 18 are students and we assign them an educational level per age (e.g., elementary school for ages between 6 and 12). For agents older than 18, we assume that either all work or that they are unemployed. We use unemployment data from the Greek Manpower Employment Organization (DYPA). We assume that all agents above 65 are retired. Then, we calculate family size using Greek Statistic Service (ELSTAT) data (https://www.statistics.gr/documents/20181/1210503/A1602_SAM01_DT_DC_00_2011_03_F_GR.pdf/\e1ac0b1c-8372-4886-acb8-d00a5a68aabe, accessed on 31 December 2023). Agent attributes are shown in Table 2.

**Table 2.** Agent attributes.

| Attribute | Description |
| --- | --- |
| Agent_ID | Agent's ID |
| Gender | Agent's gender (Male/Female) |
| Age | Agent's age from 0 to 100 |
| Family_size | Members of agent's family |
| Family_ID | Determines families |
| Work_ID | Determines workplaces, if applicable |
| School_ID | Determines schools, if applicable |
| Infection Status | Suspectible/Exposed/Infected/Recovered |

The creation of the synthetic population is based on real data from the city of Patras and consists of the following steps:

1. **Collection and preprocessing of the data:** We collected the real data on the demographics, education, employment, and other characteristics of the population from the Greek Statistic Service (ELSTAT) and the Greek Manpower Employment Organization (DYPA). These data are cleaned by removing missing and irrelevant values in order to ensure that they are accurate and consistent;

2. **Definition of agent attributes:** Based on the data collected in the previous step and the assumptions about the characteristics of the population, we defined the attributes

of the agents in the synthetic population, as shown in Table 2. We use a Python dictionary to store the attributes of each agent;

3. **Generation of the synthetic data:** We used statistical model distributions to generate synthetic data. These data represent the characteristics of the individual members of the population. These are based on the real aggregated data and the assumptions about the distribution of these characteristics;.

4. **Assignment of the attributes to the agents:** We iterated over the agents, and assigned the generated attributes to each agent, based on the synthetic data from the previous step. In other words, we "expanded" the aggregated data to agents representing members of the population by simulating the real distributions;

5. **Calculation and initialization of additional attributes:** We calculated any additional attributes that were not included in the real data, such as the infection status of each agent. This was done by randomly initializing a small number of the agents as "Infected" and the rest as "Susceptible".

The synthetic population models the population of the center of Patras city, based on the latest census by the Greek Statistical Service. We chose to model only the center of the city because of the homogeneity and statistical similarities in the population/agents. The population is organized in a multi-layer network. The layers of the network correspond to families, schools, workplaces, and age-based random interactions and we create at each step of the execution of the model a single layer that contains the random interactions.

We hypothesize family sizes from 1 to 10 members and distribute according to the distribution of the Greek population. All agents are assigned a family. If they are students, they are assigned a school and if they work, they are assigned a workplace. We assume that workplaces have 1 to 25 employees. This models the mean number of contacts in the workplace and not its total size. We distribute the employees to each workplace (Work_ID) with a Poisson distribution [56,57] since we could not find statistical data from public sources.

As for students, we place them based on their age in nursery, kindergarten, elementary, and high school. All eligible students attend school. Data from the Ministry of Education provide the number of existing schools per every level of education. The number of students for every school is produced by dividing the total number of available students by the number of schools. Then, we allocate the students to schools.

*3.2. Propagation Model*

We implement an SEIR infection model (Susceptible–Exposed–Infected–Recovered), which is an extension of the classic textbook SIR model [58]. SEIR models are extensively used in epidemiological research [59]. A population of size $N$ is partitioned into compartments that contain individuals who are Susceptible ($S$), Exposed (infected but not yet infectious), Infectious ($I$), or Recovered ($R$) [60,61]. This is a simple model that assumes no changes in the population (deaths or births), no immunity or vaccination, and no measures (e.g., obligatory usage of masks, lockdowns, etc.). In addition, due to the limited time horizon of the simulation (a small number of months), we assume that once an agent becomes recovered, they will never become infected again. There is a useful comparison of epidemiological models for transmission of SARS-CoV-2 in [62,63]. The parameters beta ($\beta$), sigma ($\sigma$), and gamma ($\gamma$), are key factors in the SEIR model [59]:

1. $\beta$: This is the transmission rate or contact rate, representing the probability of transmitting the disease from an infectious individual to a susceptible one. A higher $\beta$ generally leads to a faster spread of the disease;

2. $\sigma$: This parameter represents the rate at which exposed individuals become infectious. The reciprocal of $\sigma$ is the mean incubation period. It accounts for the time between exposure to the virus and the individual becoming infectious;

3. $\gamma$: This is the recovery rate, indicating the fraction of infected individuals recovering per unit of time. The reciprocal of $\gamma$ is the average infectious period. A shorter infectious period corresponds to a higher recovery rate.

Setting these parameters for various scenarios is a challenging task that requires objective data (e.g., new infections per day) as well as empirical information (e.g., by how much do infections reduce when masks are imposed). The main goal of the experimental evaluation at this stage is not to reproduce the historical evolution of the pandemic in Greece and consequently to make predictions, but to show trends related to the effectiveness of various policies. Some objective data for the Greek population during the pandemic can be found. However, they are not complete and require extensive manipulation to be used in order to impose restrictions on the values of these parameters. This is why we acquired values from the literature based on [64–67] and at the same time, we set the values of the parameters for the scenarios based on our subjective opinion. For the former, we used values that are suggested in the literature for similar (but not the same) propagation models, which are also tailored to particular populations, although it should be noted that these parameter values may not be correct for the Greek population. For the latter, we made an estimation based on discussions, of how much each policy affects these parameters. In this sense, the choices that we made for the various scenarios in Tables 3 and 4 are subjective and to a certain degree arbitrary. Thus, we must make clear that these values—especially the ones regarding the application of various policies—are used only to show trends related to the effectiveness of the policies, which is achieved to some degree based on the discussion in Section 4.

**Table 3.** Interventional Scenarios.

| | Scenario | Description |
|---|---|---|
| 1 | Base Case | The model runs without any interventional measures |
| 2 | School Closure | All schools are closed, all other layers remain |
| 3 | Workplace Closure | All workplaces are closed, all other layers remain |
| 4 | Targeted Age Group Interventions | All agents above 60 years old now get infected randomly with a tenth of the original possibility |
| 5 | Social Distancing & Mask usage | Possibility of Infection becomes a third of the original in all social interactions except inside families |
| 6 | Mild lockdown with (mostly) remote work | All schools are closed (tele-education), Workplace and random infection possibilities become a fourth of the original |
| 7 | Moderate lockdown with (mostly) remote work | All schools are closed (tele-education), Workplace and random infection possibilities become a sixth of the original |
| 8 | Strict lockdown with (mostly) remote work | All schools are closed (tele-education), Workplace and random infection possibilities become an eighth of the original |

**Table 4.** Parameters used in various scenarios based on [64–66].

| | $\beta\_FMLY$ | $\beta\_WORK$ | $\beta\_SCH$ | $\beta\_RNDM$ | $\beta\_SAME\_AGE$ | $\gamma$ | $\sigma$ |
|---|---|---|---|---|---|---|---|
| 1 | 0.8 | 0.1 | 0.04 | 0.01 | 0.005 | 0.1 | 0.2 |
| 2 | 0.8 | 0.1 | 0 | 0.01 | 0.005 | 0.1 | 0.2 |
| 3 | 0.8 | 0 | 0.04 | 0.01 | 0.005 | 0.1 | 0.2 |
| 4 | 0.8 | 0.1 | 0.04 | 0.01 (<60 y) | 0.005 | 0.1 | 0.2 |
| | 0.8 | 0.1 | 0.04 | 0.01/10 (>60 y) | 0.005/10 | 0.1 | 0.2 |
| 5 | 0.8 | 0.1/3 | 0.04/3 | 0.01/3 | 0.005/3 | 0.1 | 0.2 |
| 6 | 0.8 | 0.1/4 | 0 | 0.01/4 | 0.005/4 | 0.1 | 0.2 |
| 7 | 0.8 | 0.1/6 | 0 | 0.01/6 | 0.005/6 | 0.1 | 0.2 |
| 8 | 0.8 | 0.1/8 | 0 | 0.01/8 | 0.005/8 | 0.1 | 0.2 |

*3.3. Technical Description of the System*

The simulation model utilized in this study, employs a discrete-time approach. Each time step (or tick) corresponds to a single day. Our simulation involves from 60 up to 180 time steps, depending on the flatness of the curves. This corresponds to a total simulation time of 180 days (6 months) at maximum. Our reason for choosing SEIR instead of a SEIS model, where an agent can become infectious multiple times, is exactly because the number of reinfections is assumed to be near zero for a six-month period (we assume temporary immunity), although such cases exist [68–70].

The synthetic population generator outputs a CSV file, where each line of the table represents an agent with their characteristic properties: age, gender, family_id, school_id, and work_id if applicable. The synthetic generator also creates the family, school, and work networks layers by appropriately filling the aforementioned fields. Agents sharing the same family_id, school_id, or work_id belong to the same group and interact with each other accordingly.

The synthetic population of the agents is stored in a pandas dataframe, since we use the Python/Mesa ABM framework [71]. We add a column "Infection_Status" for each agent to the table, with values from the set $\{S, E, I, R\}$, representing the states of the SEIR model. Other than the three aforementioned networks, the random networks (random encounters network and random interactions based on age) are not explicitly stored, but simulated via random interactions between Infectious and Susceptible agents. We start before the execution of the model with all agents being Susceptible and by randomly infecting a small number by turning their status to Infectious. The number of initial infections is 5 (see [72] for a related discussion for the number of initial infections). No other infectious agents are injected into the system.

In each time step, we filter the Infectious agents by scanning the table of agents. Then, each Infectious agent "attempts" (with a defined probability per layer) to infect all the Susceptible agents that are in the same family, school, or workplace, or through a random interaction. In these cases, the status of the Susceptible agent turns to Exposed. The probability of infection varies in each network. For example, it is much easier to infect a member of an agent's family due to proximity and many interactions compared to the easiness of infecting someone within the workplace. An Infectious agent can infect a random agent with a very small probability. Additionally, agents with similar ages (based on the daily number of contacts from [36]) also infect one another with a small probability. The number of contacts for the age-based random layer was created by taking these probabilities from the Italian network [36]. After this infection process, for each one of the Infectious agents, it is stochastically decided as to whether each such agent becomes Recovered, and then for each one of the Exposed agents, it is stochastically decided as to whether each such agent will become Infectious. See Table 4 for the probabilities of transition between states in various scenarios. The infected agents list is updated at the beginning of each time step to include newly Infected and remove Recovered agents, filtered from the dataframe.

It is noticeable that the time needed to complete a step ("tick") increases, especially when there are many Infectious agents as the model progresses. When there are not so many Infectious agents, the simulation is fast, especially in the beginning or towards the end of the execution of the model.

## 4. Experimental Results

We ran a series of experiments to validate our model. Our experiments focused on some interventional measures. These scenarios are presented in Table 3 and the comparative results in Figures 2 and 3:

This research attempts to illustrate Level 1 empirical relevance, as defined by [73], through our model's qualitative agreement with real-world macrostructures and data. Specifically, our model aims to capture key elements of Greece's experiences during the

COVID-19 pandemic, such as population distribution and demographics, social interactions, and intervention measures.

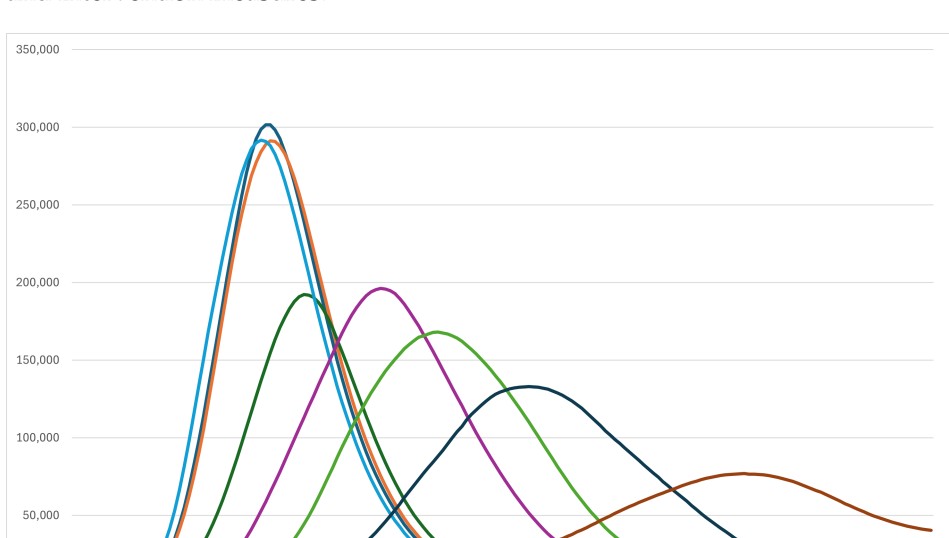

**Figure 2.** Comparison of virus spread (infections per step) in different scenarios (180 steps).

*Discussion*

Our study provides insights into the behavior and characteristics of the synthetic population we created, highlighting its implications for policy analysis and decision making. Through meticulous adherence to the steps of collecting and preprocessing real data, defining agent attributes, generating synthetic data, assigning attributes to agents, and calculating additional attributes, we successfully developed a synthetic population that carries the most basic characteristics of the population of the Greek city of Patras, Greece.

The simulation studies aimed to assess how different intervention scenarios impact the spread of COVID-19 among the Greek population. Each scenario represents a unique set of measures designed to slow down the transmission of the virus. Over the course of 100 time steps, these experiments provide insights into the effectiveness of each intervention approach. In the absence of any intervention (Scenario 1), the model depicts a rapid increase in infections, acting as a baseline for comparison. Closing all schools (Scenario 2) results in a small but noticeable decrease in infections and a delay in the peak, especially for younger students. Similarly, closing all workplaces (Scenario 3) leads to a drop in infections, highlighting the impact of workplace interactions on transmission dynamics. Tailoring interventions for individuals over 60 (Scenario 4) show a marginal effect in flattening the curve. All these scenarios emphasize the rather small impact that these policies have when applied separately. The combination of social distancing and mask usage (Scenario 5) results in a substantial decline in infections, showcasing the effectiveness of these measures in lowering the overall transmission rate. The most aggressive response, a strict lockdown with remote work (Scenarios 6–8), leads to the greatest reduction in infections, highlighting the significance of a multifaceted strategy that includes restricting social interactions, implementing remote work, and educational initiatives. Because these are stochastic models, all scenarios were run for 10 iterations and then we took averages to reduce the effect of stochasticity. Figure 3 shows a graphical comparison of these scenarios.

The multi-layer agent-based model was built using the Python/Mesa [71] framework.

Comparatively, scenarios involving school and workplace closures, along with targeted interventions for specific age groups, prove effective in controlling the virus. The combined use of masks and social distancing emerges as a potent tactic, underscoring the role of individual behaviors in limiting transmission, while the strict lockdown sce-

nario is highly effective, policymakers must carefully consider its societal and economic implications, emphasizing the need for a balanced approach.

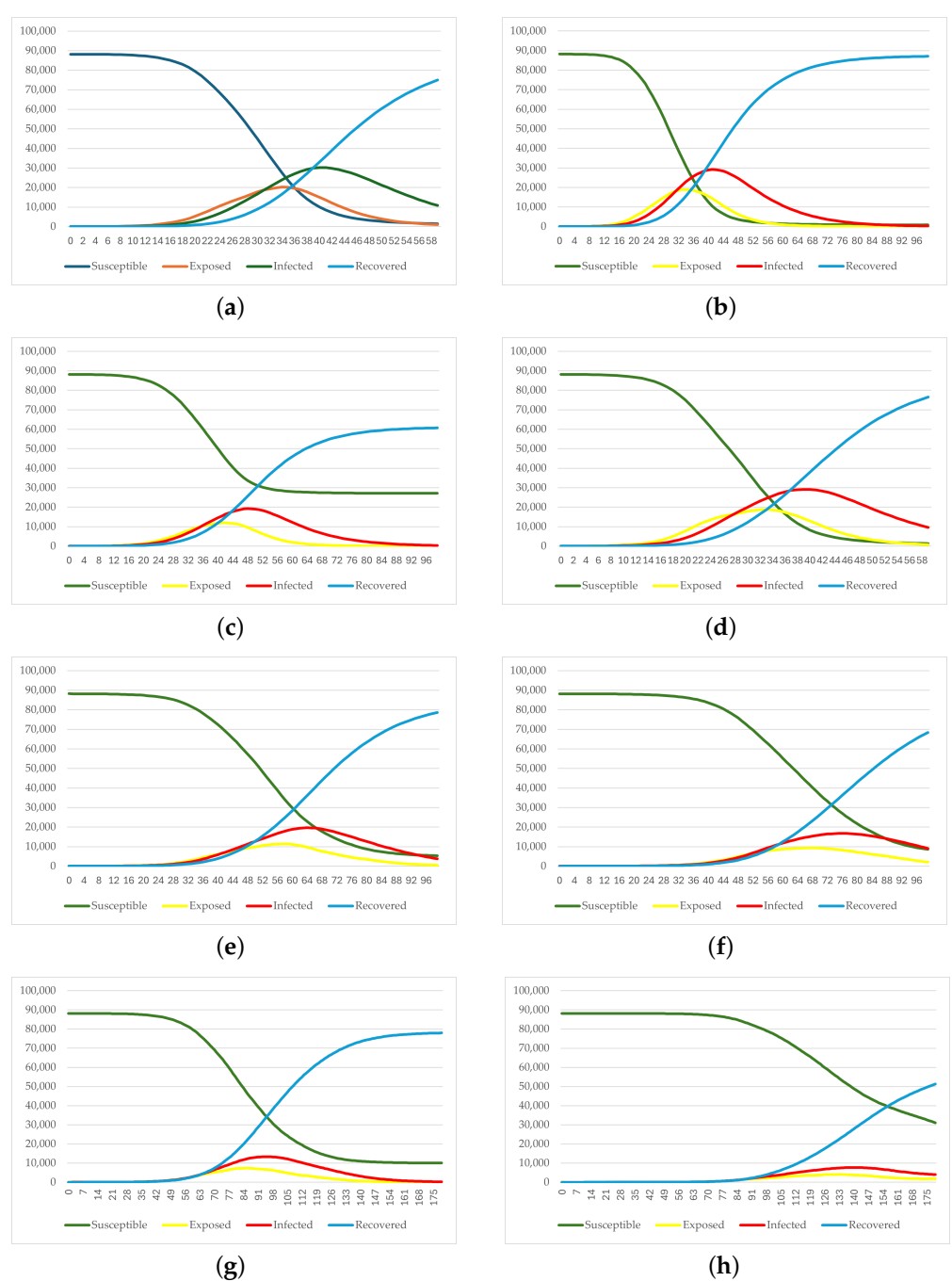

**Figure 3.** The progress in 60 steps/days of virus propagation in various scenarios. (**a**) The model runs without any interventional measures (60 steps). (**b**) All schools are closed, all other layers remain (100 steps). (**c**) All workplaces are closed, all other layers remain (100 steps). (**d**) Quarantine measures for all agents above 60 years old (60 steps). (**e**) Possibility of Infection becomes a third of the original in all social interactions except inside families (100 steps). (**f**) All schools are closed, workplace and random infection possibilities become a fourth of the original (100 steps). (**g**) All schools are closed, workplace and random infection possibilities become a sixth of the original (180 steps). (**h**) All schools are closed, workplace and random infection possibilities become an eighth of the original (180 steps).

It is crucial to note that the model's accuracy relies on precise parameterization and assumptions, and thus, real-world variations may occur. Individual compliance with inter-

ventions may vary, while the simulations assume homogeneous agent behavior. Changes in the impact of interventions may occur with new variations or shifts in human behavior over time. These simulation results provide policymakers with valuable insights for evidence-based decision making. Tailored interventions that consider societal norms and demographic variables could offer a more focused and sustainable strategy.

## 5. Conclusions

Our study demonstrates the value of synthetic populations in understanding population behavior and evaluating policy interventions. The findings from our analysis provide valuable insights for policy analysis and decision-making processes. However, it is crucial to interpret the results cautiously and consider the limitations of the ABM approach. By addressing these considerations and continuously advancing synthetic population modeling, we can contribute to evidence-based policymaking and enhance the well-being of communities and societies.

The results presented in this paper are preliminary and constitute a first step. Our next steps are the following:

1. Generalize our model to the whole Greek territory. In addition, we want to add more behaviors/traits in the population based on census and other publicly available data;
2. Extend the propagation model to take into account various states for the agents. This will make the model more realistic but at the same time will make it harder to tune since the number of parameters will increase;
3. Contact epidemiological research teams within Greece in order to further advance and tune the model and the system based on experts' opinions;
4. Extend the generator of the synthetic population towards other goals (e.g., transportation studies);
5. Extend the model to look at social implications of interventions, like economic implications, e.g., what is the economic cost of imposing an upper bound on the number of customers as a function of the area of a shop?

We also intend to use methods for data-driven parameter calibration (e.g., based on machine learning methods [74]) and rudimentary sensitivity/parameter analysis. Finally, efficiency issues will have to be considered and mitigated. In particular, to accomplish the aforementioned goals, we require the execution of multiple scenarios on large populations with complicated propagation models. Parallelism or distributed computing can greatly enhance the efficiency of the ABM approach and we intend to use ABM libraries that offer these computational modes.

**Author Contributions:** Conceptualization, K.T. and V.T.; methodology, K.T.; software, V.T.; validation, K.T. and V.T.; formal analysis, V.T.; investigation, K.T.; resources, V.T.; data curation, V.T.; writing—original draft preparation, V.T.; writing—review and editing, K.T.; visualization, V.T.; supervision, K.T.; project administration, V.T. All authors have read and agreed to the published version of the manuscript.

**Funding:** This research received no external funding.

**Institutional Review Board Statement:** Not applicable.

**Informed Consent Statement:** Not applicable.

**Data Availability Statement:** Source code and reported data can be found here: https://github.com/vorlon83/abm-patras-greece.git (accessed on 31 December 2023).

**Acknowledgments:** We would like to thank Spyros Sioutas and Theodoros Petrakis for their valuable support.

**Conflicts of Interest:** The authors declare no conflicts of interest.

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
