# Peer review of "An Agent-Based Model for Disease Epidemics in Greece"

_information, doi:10.3390/info15030150_

Round 1
Reviewer 1 Report (Previous Reviewer 1)
Comments and Suggestions for Authors
First of all, I very much want to thank the authors for the changes they made. I think the new version of the manuscript is much better (and I really appreciated the additional details on the sociocultural similarities between Italy and Greece).
I have two basic suggestions. The first is truly a suggestion and comes from my 20+ years of building agent-based models: If (and I do mean "if") one of the points of the article is to enable replication, then I would suggest following Grimm's ODD protocol to discuss the model, or better yet Mueller's ODD+D protocol. But if you're not trying to convey enough information to the reader for replication, then what's in there is fine now.
Second, you've crossed out "validate" and left in "verify." I continue to think that you don't mean verify (built the model correctly). I say that because you don't provide a detailed formulation for the model and then cross-walk between the formulation and the code base. Rather, I think you mean validate (built the correct model), meaning it relates to some referent in some non-trivial way. I'd recommend taking a look at the Axtell paper "Empirically Relevant Agent-Based Models" and the Axtell et al. paper "Aligning Simulation Models." You can use those two papers to craft a statement about what validation means in your context. I'd offer you are attempting to creating an agent-based model that has Level 1 Empirical relevance and Relational equivalence to the referent (here that referent is Greece's experiences during the COVID pandemic), you then can go on to use your results to demonstrate that you did that. This would help to provide a "punch line" to your paper, give it a sense of closure and a "so what." Given your stated interest in policy impact, doing something like this will also allow you to present your work in a context that will help policy makers understand it and how they might make use of it.
Author Response
Thank you for your kind comments.
- We are aware of ODD and ODD+D protocols to convey our ABM. But the focus on this paper was to provide preliminary results and identify challenges towards implementing a full ABM system for epidemics in Greece. In the next versions of the system we will utilize ODD+D protocol to ensure replication from the scientific community.
- We appreciate the reviewer's suggestion. We corrected "validation" instead of "verification". We have also incorporated the suggested Axtell's "Empirically Relevant ABMs" into the manuscript to underline the Level 1 empirical relevance of our ABM.
The authors

Reviewer 2 Report (Previous Reviewer 2)
Comments and Suggestions for Authors
Dear Authors. Thank you for your revisions.
Comments on the Quality of English Languageok
Author Response
Thank you for your kind comments.

Round 2
Reviewer 1 Report (Previous Reviewer 1)
Comments and Suggestions for Authors
Okay, glad the Axtell references were useful.
Comments on the Quality of English LanguageThere are a few minor typos that can easily be corrected with a final read through.
This manuscript is a resubmission of an earlier submission. The following is a list of the peer review reports and author responses from that submission.
Round 1
Reviewer 1 Report
Comments and Suggestions for Authors
This was a well written manuscript, easy to read, with good flow (save the first paragraph of section 3.1, I think that could be moved up as an introduction to the synthetic pop discussion, where it is currently reads awkwardly).
There are some issues with the manuscript in its current form, however.
1) On page two, line 83, the authors state that traditional methods disregard human interaction. I feel that is not accurate. Rather, traditional methods impose very strong assumptions about human interaction (perfect mixing).
2) The rather lengthy discussion of related studies, while a nice overview, I feel would be better used as a stand alone survey paper, and in the current manuscript shortened to highlight commonalities/unique feature of them and stress why Greece is a special case not covered by any of the prior work. Speaking of which, the absence of the "Oxford model" is striking. It was used by many nations to model pandemic dynamics, is open source, data is available, etc, so I think it meets the authors' stated requirements for inclusion. Also missing is the work of Argonne National Lab/Repast, though it received less visibility than the Oxford model. The fact that these (and others) were excluded could make perfect sense, but perhaps a bit more detail about who is in and who is out would help the reader understand.
3) (super minor) page 9, line 39, the word "agents" seems to be repeated unnecessarily.
4) the final paragraph on page 9 discusses the independence assumption and seems to say that the layers are not independent (I totally agree) but then the authors indicate that the independence assumption was used. That's fine, modeling is all about making a lot of choices, but this one seems big enough that another sentence justifying the decision is warranted. This could be as simple as "we couldn't find any data to use to figure out how the layers might be correlated" or something else....
5) Figure 1 has a number of three digit numbers in the "Tasks" column. I couldn't find in the text to what these numbers refer. If they are important, it would be helpful to have a description of them somewhere (sorry if I just missed it).
6) this is probably just my ignorance but the statement "... many similarities between Greek and Italians concerning their social relations...," surprised me. For me at least, a couple more words as to what those similarities are would help a lot.
7) There is not enough information about the model in the manuscript. There is no way you can put in everything but more detail than what's there is critical. For example: What is a time step? I can guess that it is a day based upon run duration and the statement about the simulation including a small number of months but that needs to be stated explicitly. What activation regime was used for the agents? How exactly is infection handled? Given the discussion in the manuscript I can think of three different ways to implement infection that would likely change the results and would be consistent with the description in the paper. How does the duration of a time step interact with your infection sub-model? If it is a day, can agents really only infect one other agent? Did you perform multiple runs at each parameter combination? What's the impact of stochasticity on your results? Etc.
8) The first sentence of the results section states "We ran a series of experiments to verify and validate our model." That is fantastic, but then there is no discussion of either. I suspect that by now there is data available on COVID spread in Greece so some form of validation could be performed. Minimally, you could make all agents homogeneous and fully connected. That should replicate a standard SIR model, and, at least, show you the impact of your agentization of the Greek scenario. While verification is quite important, I'm not sure that this manuscript is the place for that, so I'd recommend removing discussion of verification.
9) Table 4 is really helpful but why did you choose those numbers? Some discussion about that would be helpful to the reader (and would also help to articulate why Greece is a special case). These decisions could be based upon data or some other heuristic, but it would be helpful to the reader to know.
10) I didn't find a reference to Figure 3 in the text, it does a really nice job of showing the sensitivity of your model to your parameters so I really think you should hammer that home in the text.
Comments on the Quality of English LanguageI thought the manuscript was really well done linguistically. It is very easy to read and easy to follow (but like I said above, I think the first paragraph of section 3.1 could be moved up).
Author Response
Dear reviewer,
- The first paragraph of Section 3.1 refering to Synthetic Populations has been moved to the Preliminaries
- This phrase has been corrected as per your comment.
- We have added a reference to the Oxford Model and Repast. As for the related work, the reviewer is correct that it is quite lengthy. However, we believe that the related work should be lengthy for the following reasons: a) It gives a glimpse as to how the field has proliferated the last years. b) it places our work within a particular context. c) Evolving it to a survey paper is in our future plans since the current related work is truly a glimpse of the literature. We have gathered a lot of related material that we decided not to include in the paper since this section would be at least three times its current length.
- Corrected as per your comment.
- We added a related comment in the text.
- Corrected. These values were artifacts during the transfer of the images to the latex source.
- There are strong similarities between the two neighboring Mediterrenean countries of Greece and Italy, for example: common history and cultural values, strong family ties and laid-back social atmosphere. Because we weren't able to find social interaction data for Greece, we used these data from the Italian population, given the many similarities between them. This is a common practice in research, especially when specific data for one region is not readily available, but a comparable region is deemed representative. We have added in the text a related discussion.
- We added a subsection titled "Technical Description of the System" that answers these questrions. This was a major ommission from our side and we thank the reviewer fotr bringing it up.
- We removed the reference to validation since it is was not tackled.
We added a related discussion as to how we chose these values. As highlighted in the discussion, and maybe not explicitly said in the previous version of the paper, our goal at this stage is to look at the trends and not to find a model that closely follows the evolution of the epidemic in Greece. - Added a reference to Figure 3 in the Discussion section.
Please see the attachment.
Best Regards

Reviewer 2 Report
Comments and Suggestions for Authors
Dear Authors.
Thank you for considering MDPI's information for publication. Below are some comments and questions:
1. You have given an interesting overview (2.1) of existing models to support policy decisions about Covid. What is the scientific gap that your model addresses? What makes your model unique compared to other models?
2. Unfortunately, I could not find any details about the design and realization of the model you presented. Other papers published in this magazine usually contain more technical information.
3. The results you show in Figs. 2 and 3 look very basic. I assume that your model is not deterministic. So there should be some deviations in your results. Perhaps it would be useful to show these in boxplot diagrams or something similar.
4. I have not been able to find out how you validated your model. Political decisions are closely linked to credibility and trust. How can you realize this?
Comments on the Quality of English LanguageNo comments.
Author Response
Dear reviewer,
- This model addresses a SEIR ABM disease model in a synthetic population of a major Greek town (Patras). It's novelty is that it is adapted to the pecularities of the Greek population based on publicly available data. By creating this model we aim to provide insights on the evolution of a COVID-19 pandemic (or for any other virus) in the Greek population. We added a related discussion.
- We added a subsection titled "Technical Description of the System" that answers these questrions. This was a major ommission from our side and we thank the reviewer fotr bringing it up.
- Because this is a stochastic process, we ran 10 iterations for each scenario and then calculated averages. All graphs are based on these averages. We also added a related comment.
- We removed the reference to validation since it is was not tackled.
Please see the attachment.
Best Regards
